# The Efficiency and Safety of Platelet-Rich Plasma Dressing in the Treatment of Chronic Wounds: A Systematic Review and Meta-Analysis of Randomized Controlled Trials

**DOI:** 10.3390/jpm13030430

**Published:** 2023-02-27

**Authors:** Shang Li, Fei Xing, Tongtong Yan, Siya Zhang, Fengchao Chen

**Affiliations:** 1Medical Cosmetic Center, Beijing Friendship Hospital, Capital Medical University, Beijing 100050, China; 2Department of Orthopedics, Orthopedic Research Institute, West China Hospital, Sichuan University, Chengdu 610041, China

**Keywords:** wound healing, platelet-rich plasma, chronic ulcers, chronic wounds, wound repair

## Abstract

Recently, many clinical trials have applied platelet-rich plasma (PRP) dressings to treat wounds that have stopped healing, which are also called chronic wounds. However, the clinical efficiency of PRP dressings in treating chronic wounds is still controversial. Therefore, we conducted this study to compare PRP dressings with normal saline dressings in treating chronic wounds. Relevant randomized controlled trials focusing on utilizing PRP dressings in treating chronic wounds were extracted from bibliographic databases. Finally, 330 patients with chronic wounds, reported in eight randomized controlled trials, were included in this study. In total, 169 out of 330 (51.21%) were treated with PRP dressings, and 161 out of 330 (48.79%) were treated with normal saline dressings. The pooled results showed that the complete healing rate of the PRP group was significantly higher than that of saline group at 8 weeks and 12 weeks, respectively. In addition, there were no significant differences in wound infection and adverse events. Compared with normal saline dressing, the PRP dressing could effectively enhance the prognosis of chronic wounds. Furthermore, the PRP did not increase wound infection rate or occurrence of adverse events as an available treatment for chronic wounds.

## 1. Introduction

Chronic wounds constitute a common and refractory disease in the surgical department, and present poor healing or non-union during long-duration observation. The pathogenesis of chronic wounds is complicated and indistinct, including inflammation [1], chronic disease, infection, angiogenesis disorders, malnutrition, aging, dystrophy, local pressure, and oedema [2]. In histology, the local tissues of chronic wounds show a large amount of neutrophil infiltration which leads to excessive inflammation around the wound and further recruits reactive oxygen species (ROS) and destructive enzymes to perpetuate the cycle [3]. Chronic and excessive inflammation inhibit wound healing. In clinical treatment, chronic wounds have a high incidence rate, which could be divided into venous ulcers, arterial ulcers, pressure ulcers, and diabetic ulcers [4]. Among these classifications, the venous ulcer is the most common wound. In various wounds, macromolecules and red blood cells are exuded from vessels and gather in interstitial space, further inducing leukocyte infiltration and chronic inflammation [5]. Arterial ulcers and pressure ulcers occur because of thrombosis or physical pressure, which usually lead to local ischemia and hypoxia and activate local chronic inflammation. The diabetic ulcers are due to glycation end-products accumulating in systemic tissues and producing oxidative stress, vascular atrophy, and circulatory dysfunctions [6]. The traditional treatments include debridement, wound bed preparation, normal saline dressing, and infection management [7]. Commonly, symptomatic treatments are useless in improving local blood circulation or reducing local inflammation. Therefore, most patients suffer from long-term treatment and chronic pain, which decrease their quality of life and increase their financial burden [8,9]. Following the high incidence rate of chronic wounds, the worldwide medical system suffers a heavy economic burden, which induces a high risk of thanatosis, amputation, and death [10]. In order to improve the healing of chronic wounds, several kinds of functional dressings have also been used to treat chronic wounds in clinical treatment, such as hydrogel dressings and antimicrobial dressings. Hydrogel dressings can absorb the exudate of wounds and keep them moist, providing an appropriate environment for wound healing, but they are expensive for use in long-term treatment and lack abundant clinical evidence supporting their efficacy [11]. Iodine or silver-based dressings have proven antimicrobial ability in wound treatment, but some researchers are concerned that the high concentration of metallic ions can harm the human body, and long-term usage might delay healing [12]. There are also some biomaterial dressings used in chronic wound management, possessing biocompatibility and biodegradability and antimicrobial properties, but lacking sufficient clinical evidence [13]. Therefore, the efficacy and safety of these functional dressings are still lacking sufficient long-term clinical experiments, and they are therefore hard to widely apply in clinical treatment. Therefore, a novel treatment to promote chronic wound healing is necessary to remedy the insufficiency of traditional treatment [14].

Various studies have found that growth factors and cytokines play irreplaceable roles in modulating tissue repair and regeneration, especially in bone, skin, cartilage, and vascularized tissues [15]. Platelet-rich plasma (PRP) has been extracted from the peripheral blood of patients and showed a concentration of various growth factors and cytokines without known adverse effects [16,17]. As we all know, PRP could release various biologically active factors and adhesion proteins into the microenvironment, which may contribute to initiating hemostatic cascade, vascularization, and tissue regeneration [18,19]. Many studies have confirmed that growth factors derived from PRP are able to shorten the wound healing process via the supraphysiological releasing of growth factors promoting cell proliferation, migration, and vascularization [20]. Neovascularization is an essential process to reconstruct blood supply and support the high metabolic activity of tissue regeneration. In animal experiments, it has been found that senescent stem cells could recover proliferation and colony formation ability after PRP treatment, confirming that PRP could resist cell senescence during tissue regeneration [21]. Besides, when PRP is exposed to wounds, the platelets, the main ingredients in PRP, are released as platelet lysate by freeze-thawing disruption [22,23]. The platelet aggregation is activated, and they lead to a cascade reaction of cytokines, producing an amount of pain-modulating 5-hydroxytryptamine sufficient to relieve local pain [24]. These mechanisms have encouraged many clinical studies concentrating on PRP dressings to treat chronic wounds [25,26]. In many studies, PRP dressings have been applied to chronic wounds, compared with saline dressing. Some studies have presented satisfactory results using PRP dressings; however, some studies have presented conflicting outcomes [27]. Therefore, we conducted this study to explore the healing rate and complications of PRP dressings in treating chronic wounds.

## 2. Materials and Methods

The study was conducted according to Preferred Reporting Items for Systematic reviews and Meta-Analysis (PRISMA) guidelines [28]. The study included objectives, literature research, eligibility criteria, data extraction, outcome measures, an assessment of methodological quality, and statistical analyses.

### 2.1. Literature Research

Several potentially relevant works published in electronic databases were independently searched by two reviewers, including Embase, Web of Science, PubMed, Medline, and Cochrane Central Register of Controlled Trials. The keywords used in the process of searching were as follows: “autologous plasma,” “PRP,” “Platelet-rich plasma,” “chronic wound,” “chronic wounds” “nonhealing wound,” “wound,” “chronic ulcer,” “nonhealing ulcer,” and “ulcer.” The studies focusing on PRP dressings used in the treatment of chronic wounds were included. The researchers initially evaluated the titles and abstracts of the search results. After that, the full texts were reviewed thoroughly to extract information. In addition, in order to maximize the search results, the researchers also reviewed the references of the retrieved studies to search for more relevant studies in the process of literature research.

### 2.2. Eligibility Criteria

The inclusion criteria for these studies were as follows: (1) studies focusing on patients suffering from chronic wounds which lasted at least four weeks without healing; (2) prospective randomized control studies; (3) studies in which the experimental group comprised of the administration of a PRP dressing combined with wound cleaning or debridement, with a control group involving normal saline dressing with wound cleaning or debridement combined; (4) studies showing PRP preparation procedures; And (5) articles which were written in English.

The exclusion criteria for these studies were as follows: (1) those investigating wounds lasting less than four weeks or which had no description of the time; (2) those studies with experiment groups involving the administration of PRP injections; (3) retrospective control studies and retrospective cohort studies; (4) comment papers; (5) case reports; (6) protocol descriptions; and (7) those studies without specified outcomes.

### 2.3. Data Extraction

The data of patients were extracted by two reviewers independently. The demographic characteristics extracted included the medical center, first author, publishing year, the sample size in each study, the average age of patients, follow-ups, inclusion criteria, and exclusion criteria. The interventional factors extracted included types of wounds, wound area, experiment treatment, control treatment, and treatment duration. If there were disputes between two reviewers, they were settled through consultation with a third reviewer.

### 2.4. Outcome Measure

The clinical outcomes included the complete healing rate at twelve weeks, the infection rate, and the adverse events. The clinical healing rate and complications of PRP dressings used to treat chronic wounds were evaluated.

### 2.5. Assessment of Methodological Quality

The methodological quality of randomized controlled studies in this study was assessed by two reviewers independently. In addition, the bias of randomized comparative studies was also assessed by modified Jadad scores [29]. When the modified Jadad scores were ≥4 points from a possible total of 8, the studies were considered high quality.

### 2.6. Statistical Analysis

Two reviewers independently conducted a statistical analysis using RevMan Manager 5.3. The mean differences (MDs) with a 95% confidence interval (95% CI) were used to evaluate continuous variables. The risk ratio (RR) or risk difference (RD) with a 95% confidence interval (95% CI) was used to evaluate dichotomous data. *p* < 0.05 was considered statistically significant. The weighted mean difference (WMD) between groups of PRP and controls was reported with 95% confidence intervals (95% CI). Q chi-square test and I^2^ statistic were used to evaluate the statistical heterogeneity for all enrolled studies. In addition, when I^2^ was >50%, the randomized-effects model was used. Otherwise, the fixed-effects model was chosen.

## 3. Results

### 3.1. Study Selection

A total of 103 relevant publications were retrieved after a literature search. Sixteen studies were excluded because of duplicate records. Then, 87 studies were screened through their titles and abstracts. At last, 8 randomized controlled trials (RCTs), comprising 330 patients with chronic wounds, met the selection criteria and were selected for inclusion in this study [30,31,32,33,34,35,36,37]. Figure 1 shows the flow diagram of this systematic review.

### 3.2. Study Characteristics

The demographic characteristics of the used studies were extracted and are shown in Table 1. In total, 330 patients, including 201 males and 129 females, were represented in the present study. All studies reported the administration of PRP dressings in treating chronic wounds. The sample size ranged from 13 to 102. In addition, the presentation time of chronic wounds ranged from 4 weeks to 6 months. The follow-up ranged from 1.5 to 6 months. Furthermore, among these RCTs, four were performed in Egypt, two were performed in Spain, and one each in France and America.

The types of chronic wounds and treatment duration in all investigated studies are presented in Table 2. The mean wound area in these studies ranged from 3.4 ± 4.5 to 33.70 ± 53.32 cm^2^ in the PRP groups and 2.64 ± 0.48 to 16.67 ± 23.87 cm^2^ in the normal saline groups. The types of wounds in all investigated studies included chronic venous ulcers, diabetic foot ulcers, and chronic cutaneous ulcers. The treatment duration in these studies ranged from 6 weeks to 20 weeks.

### 3.3. Risk of Bias

The risk of bias in the investigated RCTs was evaluated by two reviewers separately. Table 3 shows the modified Jadad scores of the included RCTs. The mean of the modified Jadad score of all RCTs was 4.5 (range, 2–7). Additionally, Figure 2 shows the risk of bias in all prospective clinical studies.

### 3.4. The Complete Healing Rate

Two RCTs reported complete healing rates at 8 weeks in both the PRP and normal saline dressing groups. The pooled results found that the healing rates of PRP dressings was significantly higher than those of the normal saline dressing groups at 8 weeks (OR = 11.70; 95%CI, 1.40 to 97.97; *p* = 0.02). There was no significant heterogeneity between the two types of groups (*p* = 0.37, I^2^ = 0%) (Figure 3A). Three RCTs reported complete healing rates at 12 weeks in both the PRP groups and the normal saline groups. The pooled results confirmed that the healing rates of the PRP groups were significantly higher than those of the normal saline groups at 12 weeks (OR = 6.56; 95%CI, 1.09 to 39.47; *p* = 0.04). There was significant heterogeneity between the two types of groups (*p* = 0.03, I^2^ = 70%) (Figure 3B).

### 3.5. Infection Rate and Adverse Events

Four RCTs reported infection rates during treatment of both the PRP dressing groups and the normal saline dressing groups. In addition, no significant differences were found in infection rates between the PRP and normal saline dressing groups (OR = 0.53; 95%CI, 0.20 to 1.44; *p* = 0.21). No significant heterogeneity was found in the infection rate between the PRP and normal saline dressing groups (*p* = 0.57, I^2^ = 0%) (Figure 4A). Six RCTs reported the rate of adverse events in both PRP group and normal saline group. The adverse events included infection, thrombophlebitis, irritative dermatitis, and death. There was no significant difference in the rate of adverse events between the PRP groups and normal saline groups (OR = 0.61; 95%CI, 0.25 to 1.45; *p* = 0.26). No significant heterogeneity was found in the infection rate between the PRP and normal saline dressing groups (*p* = 0.42, I^2^ = 0%) (Figure 4B).

## 4. Discussion

This systematic review compared PRP and normal saline dressings in treating chronic wounds. There were eight RCTs investigated in the present study, and the results confirmed that PRP dressings could effectively enhance the healing rate of chronic wounds after 8 weeks and 12 weeks of treatment compared with normal saline dressings. Further studies found that the PRP dressing was safe for treating chronic wounds without raising the infection rate and adverse events rate.

Chronic wounds typically present after healing has stopped, resulting in high morbidity and mortality rates around the world [38,39]. The pathology of chronic wounds is complicated, including diabetes, venous disease, and arterial disease, leading to the blocking of blood and energy support, and further resulting in ischemia and hypoxia. The local ROS and inflammation are activated by ischemic and hypoxic tissue, finally slowing down the healing progress. Besides the damage caused by local inflammation, wound infection, such as polymicrobial infecting and biofilm formation, further disrupts the healing process [40,41]. During normal cutaneous healing, inflammation is actually a necessary process to eliminate pathogenic microorganisms and senescent cells and promote tissue regeneration. However, in chronic wounds, the long-term stimulation of ischemic and necrotic tissue leads to excessive inflammation and further tissue damage. Nowadays, traditional wound care involves regular wound cleaning and dressing changing [42]. Saline dressings and Vaseline gauze are the most common choices in clinical treatment because of their safety, effectivity, and universality. Saline dressings should be changed at appropriate intervals to keep them clean and dry. Hydrogel dressings comprise a widely used biomaterial in clinical treatment, which can provide a moist environment and gas exchange, protect wounds from microorganisms, and promote angiogenesis and tissue regeneration via releasing bioactive factors [43]. Most hydrogel dressings present satisfactory effectiveness in clinical treatment, though their high production cost limits their long-term usage in chronic wounds [44]. On this basis, metallic-based dressings are used in chronic wounds. Silver-based dressings are usually used in infected chronic wounds, presenting antibacterial properties, modulating inflammation, and promoting wound healing effects in clinical treatment [45]. However, there are some controversial outcomes regarding the safety of the long-term and abundant usage in the human body and the environment. Therefore, in clinical treatment, most patients with chronic wounds have low quality of life because of the repetitive treatment and chronic pain, and they also suffer from anxiety due to social isolation, interrupted sleep patterns, hopelessness, and dysfunction [46]. Therefore, finding an available treatment to promote wound healing is necessary for patients suffering from chronic wounds.

In recent years, many clinical studies have focused on chronic wound treatment, such as PRP, bone marrow mesenchymal stem cells [47,48], transcutaneous electric nerve stimulations [49], and metallic dressing [50,51,52]. Among these treatments, PRP is an advanced method because of its pleasant effect, simple procedure, low cost, and safety [25]. PRP consists of large amounts of growth factors and cytokines, which are released onto the wound and can recruit stem cells and various kinds of growth factors via cascade reactions. These cells and factors can contribute to regulating epithelial cell proliferation and migration, regulating fibroblastic activity [18], promoting angiogenesis and vessel permeability [53], and increasing protein and extracellular matrix synthesis [54]. In vivo, these growth factors could enhance the metabolic reprogramming of fibroblasts, especially promoting their glycolytic energy metabolism, to stimulate fibroblast proliferation and differentiation during tissue repairing [55]. Additionally, concentrated platelets could stimulate the proliferation and pro-angiogenic properties of mesenchymal stem cells even under oxidative stress to promote angiogenesis and metabolic support around the wound [56]. Besides, platelet accumulation can be stimulated by endothelial injury and microbial pathogens in chronic wounds, which could regulate leukocyte oxidative bursting to stimulate the immune response via platelet–neutrophil interactions [57]. The activated immune system could then quickly identify and eliminate the viruses and bacteria around chronic wounds, resulting in the anti-infection effect of PRP dressings [58]. The inflammation would be inhibited after eliminating infection, and the red granulation tissue formation would be shown at the wound site [59]. Through these mechanisms, PRP presents a promotion in tissue regeneration and chronic wound healing. Nowadays, PRP has been applied in clinical trials and is supposed to be a potential treatment for chronic wound treatment. Among these studies, there were two routes of PRP administration found: one constituted injecting PRP around the wound, and the other one constituted wrapping the wound with PRP dressing [60]. Because of the high risk of subcutaneous injection and the pain of injection, most studies focused on the clinical healing rate and complications of PRP dressings in treating chronic wounds.

In this study, we found that PRP dressings could significantly increase the healing rate of chronic wounds after 8 weeks and 12 weeks of treatment. The other included studies also reported consistent outcomes at different endpoints. In Senet et al. [30]’s study, there were no significant differences found in the healing rate every day and the complete healing rate between the two groups, while the PRP dressing group showed a trend of greater healing rate than the normal saline dressing group. Elsaid et al. [37] reported a complete healing rate at 20 weeks; the PRP dressing group was significantly higher than the normal saline dressing group (3/25 vs. 0/25, *p* = 0.03), and they showed that the time of wound to maximum healing was significantly shorter in the PRP dressing group than in the normal saline dressing group (6.3 ± 2.1 vs. 10.4 ± 1.7 weeks, *p* < 0.0001). Elsaid et al. [37] also found the inefficient rate was significantly lower in the PRP group; only 8% of wounds presented no response to PRP dressings, while more than 66% of wounds had no response to saline dressings. Moneib et al. [35] reported a complete healing rate at end of the study; the PRP group was significantly higher than the normal saline group (7/20 vs. 0/20, *p* = 0.04), and they also recorded the linear healing rate of two groups; the PRP group showed a significantly higher healing rate (*p* < 0.05). Manuel et al. [34] reported the percentage of the healed area after 24 weeks of treatment, and found that the PRP dressing group was significantly quicker than the normal saline dressing group (67.7% ± 41.54 vs. 11.17% ± 24.4, *p* = 0.001), and the PRP group had significant pain reduction (*p* = 0.001). Elgarhy et al. [36] evaluated the inflammation and regeneration of chronic wounds via histologic staining. They found that local tissues presented less inflammatory cell infiltration and well-formed granulation tissues after six weeks of PRP dressing treatment, while moderate vascular proliferation and marked chronic inflammatory cells after six weeks of saline dressing treatment. The results from these RCTs were unanimous in terms of the PRP dressing’s ability to significantly promote chronic wound healing, compared with saline dressing. In addition, some researchers found that chronic wounds in the PRP group showed bright red granulation after 4 weeks of treatment, which was helpful to reduce inflammation exudation [61]. Pires et al. [62] found that there was a similar amount of bacteriological growth in the excretion culturing of the PRP and normal saline dressing groups. These results confirmed the mechanisms of PRP, i.e., that growth factors from PRP could reduce local inflammation and promote angiogenesis and tissue regeneration around the wound.

Besides the healing rate, chronic pain was another main factor decreasing patients’ quality of life. During chronic wound management, pain reduction should be a vital clinical goal and managed as early as possible. However, wound pain was repetitive and refractory in clinical treatment, and had similar characteristics to chronic pain, such as pain centralization and long-term drug use. The mechanisms of chronic pain were complicated; some studies pointed out that the 5-hydroxytryptamine system played vital roles in the modulating magnitude of pain. Many studies tried to find the mechanisms of PRP decreasing pain, and they found PRP could produce an amount of pain-modulating 5-hydroxytryptamine to relieve local pain [24]. In the investigated RCTs, some studies reported that PRP dressing could decrease pain compared with the normal saline dressing group [34]. Patients’ satisfaction also significantly increased during treatment followed by pain relief [34,36]. While, honestly, PRP was not a strong pain inhibitor, it was hard to achieve the goal of an entirely painless wound after PRP treatment.

In clinical treatment, the infection rate and complications of PRP dressing are serious concerns to doctors. According to this study, the infection rate and adverse events had no significant differences between the PRP dressing groups and the saline dressing groups. Among the included RCTs, there was only one study which reported a case of thrombophlebitis during sampling [30], which did not lead to serious complications. Additionally, one study reported two cases of death, one each in the PRP and saline dressing groups [31]; their deaths were unrelated to the chronic wound. Two studies reported that no complications happened during treatment [36,37]. These results confirmed that PRP dressings would not increase the rate of complications, and they were safe for the regular treatment of chronic wounds.

There were still several limitations in this systematic review. The primary limitation was that the intervals of changing PRP dressing were different in these RCTs, which may enhance the bias of the outcomes. Secondly, because the procedures of changing the dressings were different in the PRP and saline dressing groups, the allocation concealment was not involved with patients or operators in some RCTs, which may enhance the bias of the outcomes. Thirdly, the number of RCTs we included was limited. The outcomes of wounds were irregular, and some data could not be used, so the follow-up durations were short at the meta-analysis stage. We reported relative outcomes of all enrolled RCTs to increase objectivity. We tried to include as many patients as possible, but the results may still have some limitations.

## 5. Conclusions

Compared with normal saline dressings, PRP dressings could significantly increase the healing rate of chronic wounds. Additionally, the PRP dressings were shown to be safe for patients, and regular treatment did not increase the infection rate and adverse events in chronic wounds. Although there were some limitations in the reported studies, the PRP dressing should serve as an available treatment for chronic wounds in the future. However, more clinical trials with large-scale, standard, and long-term characteristics need to be conducted to further investigate the effectiveness and safety of PRP dressings in treating chronic wounds.

## Figures and Tables

**Figure 1 jpm-13-00430-f001:**
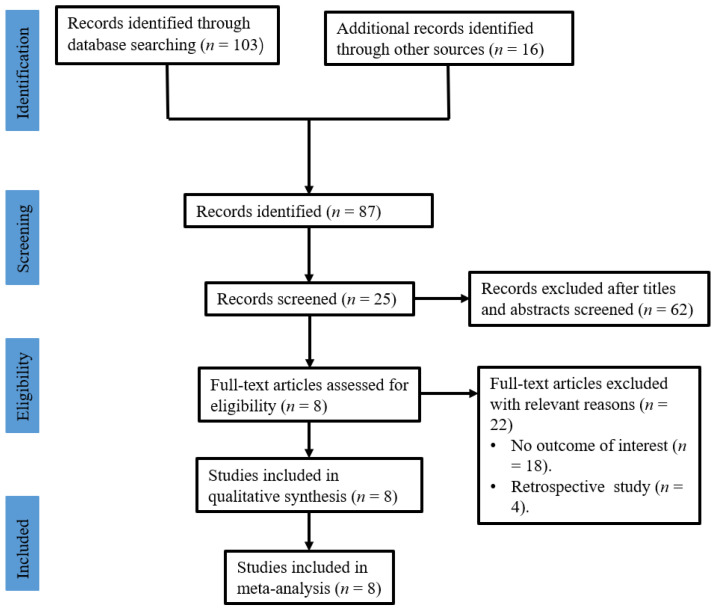
The flow diagram of this study.

**Figure 2 jpm-13-00430-f002:**
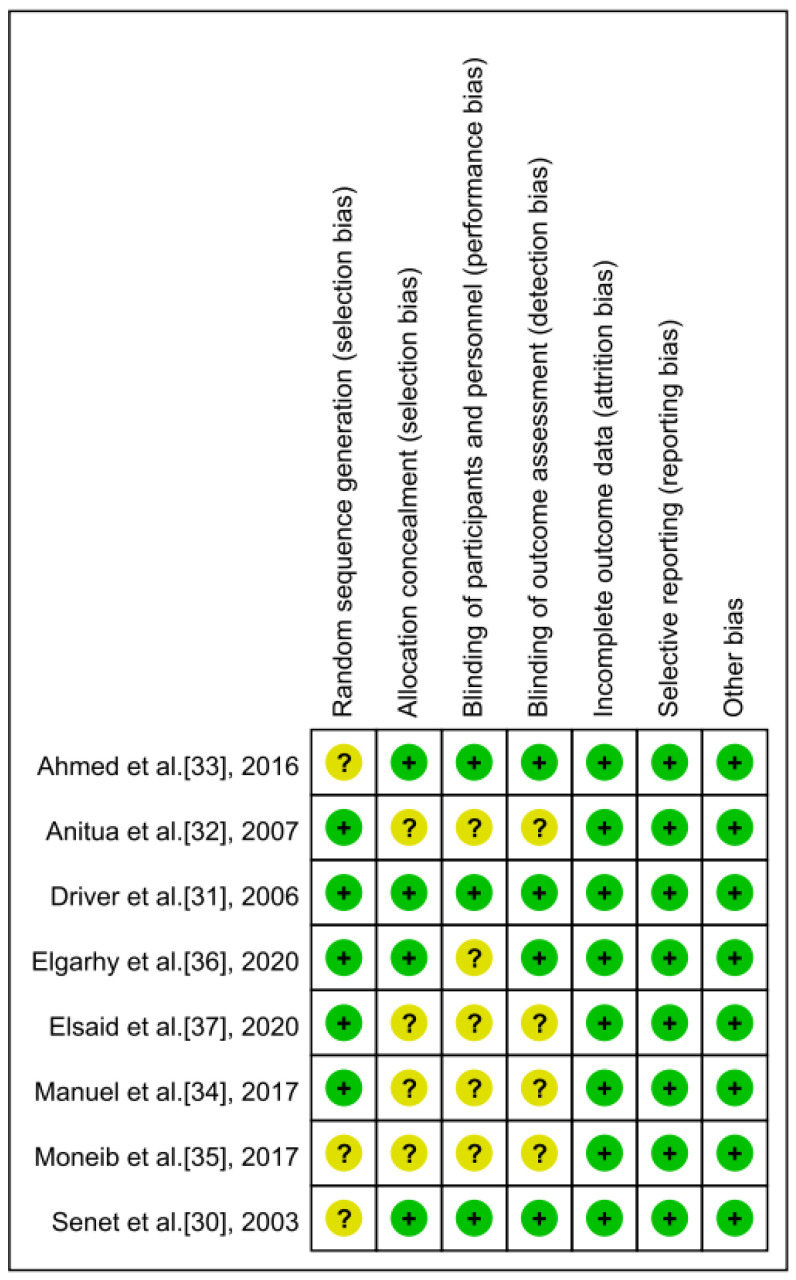
The risk of bias in all enrolled studies.

**Figure 3 jpm-13-00430-f003:**
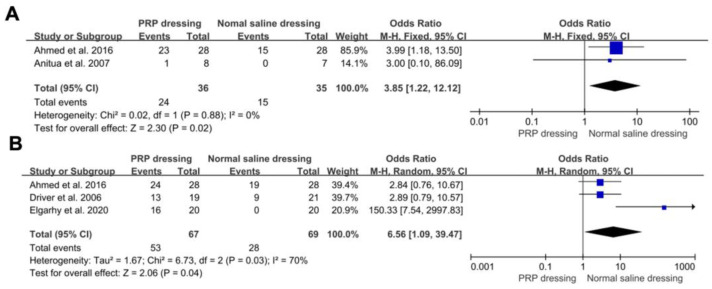
Forest plot showing the healing rates of PRP dressing versus normal saline dressing groups. (**A**) The complete healing rates of PRP and normal saline dressing groups at 8 weeks. The events were the number of complete healing wounds at 8 weeks. (**B**) The complete healing rates of PRP and normal saline dressing groups at 12 weeks. The events were the number of complete healing wounds at 12 weeks.

**Figure 4 jpm-13-00430-f004:**
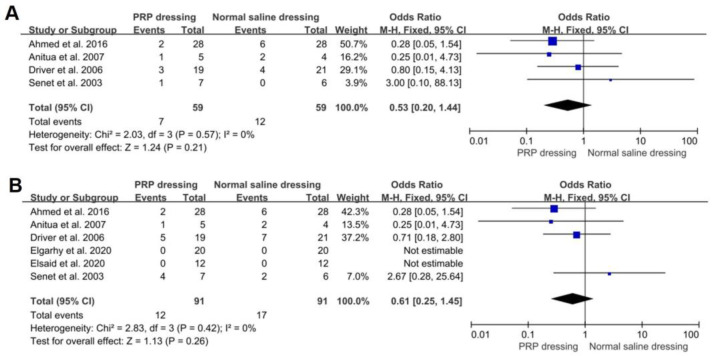
Forest plot showing the rate of complications of PRP dressing versus normal saline dressing. (**A**) Infection rate during treatment. The events were the number of infected wounds after treatment. (**B**) Adverse events rate during treatment. The events were the number of adverse events after treatment.

**Table 1 jpm-13-00430-t001:** The demographic characteristics of all investigated RCTs. P represents the PRP group. C represents the normal saline group.

Study	Country	Patients (P/C)	Age (P/C) Years	Male (P/C)	Follow-Up (Months)
Senet et al. [30], 2003	France	7/6	72.3 (45–88)/72.3 (50–83)	4/3	4
Driver et al. [31], 2006	America	19/21	58.3 ± 9.7/55.9 ± 8.1	16/16	3
Anitua et al. [32], 2007	Spain	8/7	45 ± 20/61 ± 16	4/4	2
Ahmed et al. [33], 2016	Egypt	28/28	43.2 ± 18.2/49.8 ± 15.4	20/18	3
Manuel et al. [34], 2017	Spain	55/47	65 ± 13.72/69 ± 16.26	15/16	6
Moneib et al. [35], 2017	Egypt	20/20	36.4 ± 10.2/32.5 ± 7.5	19/20	1.5
Elgarhy et al. [36], 2020	Egypt	20/20	43.70 ± 13.12/43.50 ± 8.10	16/16	3
Elsaid et al. [37], 2020	Egypt	12/12	54.7 ± 6.6/55.6 ± 6.5	8/6	5

**Table 2 jpm-13-00430-t002:** The types of chronic wounds and treatment duration in all investigated studies. P represents the PRP group. C represents the normal saline group.

Study	Type of Wound	Wound Area (P/C)/cm^2^	Treatment Duration
Senet et al. [30], 2003	Chronic venous ulcers	13.7 (4.8–27.25)/10.85 (3.7–26.5)	Three times per week until either complete healing or 12 weeks of treatment.
Driver et al. [31], 2006	Diabetic ulcers	3.4 ± 4.5/3.6 ± 4.0	Twice a week until the wound healed or a maximum of 12 weeks.
Anitua et al. [32], 2007	Chronic cutaneous ulcers	5.5 ± 4.8/8.9 ± 8.6	Until the wound healed or a maximum 8 weeks.
Ahmed et al. [33], 2016	Diabetic ulcers	6.24 ± 0.77/2.64 ± 0.48	Wound closure or occurrence of infection or a maximum of 3 months.
Manuel et al. [34], 2017	Venous ulcers	13.69 ± 30/16.67 ± 23.87	Saline cleaning once every 3 days and PRP application once a week until the wound healed.
Moneib et al. [35], 2017	Chronic venous leg ulcers	7.97 ± 16.88/2.94 ± 1.22	Once per week until the wound healed or a maximum of 6 weeks.
Elgarhy et al. [36], 2020	Chronic venous leg ulcer	33.70 ± 53.32/15.0 ± 8.30	PRP application was used weekly until the wound healed or a maximum 3 of months.
Elsaid et al. [37], 2020	Non-healing diabetic foot	-	PRP was used until the wound healed or a maximum of 20 weeks.

**Table 3 jpm-13-00430-t003:** Modified Jadad Score for the investigated studies. Each asterisk represents one point. The score is used to assess the quality of clinical trials; when the score was ≥4 points, the trials were considered high quality.

Study (Year)	Randomization	Concealment of Allocation	Double Blinding	Total Withdrawals and Dropouts	Total
Senet et al. [30], 2003	*	**	**	*	6
Driver et al. [31], 2006	**	**	**	*	7
Anitua et al. [32], 2007	**	-	-	*	3
Ahmed et al. [33], 2016	*	**	**	*	6
Manuel et al. [34], 2017	**	-	-	*	3
Moneib et al. [35], 2017	*	-	-	*	2
Elgarhy et al. [36], 2020	**	**	*	*	6
Elsaid et al. [37], 2020	**	-	-	*	3

## Data Availability

No new data were created or analyzed in this study. Data sharing is not applicable to this article.

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
