# Peer review of "The Efficiency and Safety of Platelet-Rich Plasma Dressing in the Treatment of Chronic Wounds: A Systematic Review and Meta-Analysis of Randomized Controlled Trials"

_jpm, 2023, doi:10.3390/jpm13030430_

Round 1

Reviewer 1 Report

Dear authors,

thank you for the opportunity to review your manuscript "The efficiency and safety of platelet-rich plasma dressing in treatment for chronic wounds: a systematic review and meta-analysis of randomized controlled trials", in which you conclude the superiority of PRP-dressings compared to saline dressings in chronic wounds. While I think your manuscript is a valuable contribution to the scientific community dealing with chronic wounds, there are some things to improve before I can recommend publication:

1. Your first part of the introduction is just the abstract copied in there (until line 46). I recommend to properly read your manuscript before you submit as as mistake like this rather appears careless showing a certain kind of disrespect towards editorial team and reviewers.

2. Your introduction is rather superficial. You write about growth factors and PRP but on a very superficial basis. Given that the journal is not a specific journal about these two (and even then), I would prefer to read more about current standards, postulated working mechanisms, other indications for the use of PRP and so on.

3. Furthermore, you write that traditional treatment included ... the use of "normal saline dressing"... While this might be true to some extent, many new and other wound dressings have emerged that could be used effectively in chronic wounds (some more and some less) in recent years and decades. I recommend to screen the literature again to show some examples and to introduce new insights in wound healing, e.g. about pH and lactate.

4. In the methods section: I do not understand how you retrieved unpublished studies (line 83) - are you talking about own case series? Or non-reviewed studies published online? Either way, I recommend to NOT include these in your review as this significantly dilutes your results lowering the quality. Same goes for "eligible studies were searched via Google search engine" as Google search (not even google scholar) is not an appropriate means to find sound and peer-reviewed scientific information.

5. Have you registered your review on PROSPERO? If so, please indicate.

6. You report no heterogeneity between studies reporting a 12-week outcome (line 188), yet the I2 reaches 70% - in my understanding, this heterogeneity is to be considered rather high...? Please comment on that.

7. Two RCTs reported the healing rate at 8 weeks, and three at 12 weeks... What about the other 3? Did they not report any healing rate at all? If so, please state - I would also like to know the pooled healing rate of all studies. Since this is the main outcome of your study, I think this is of high relevance.

8. Please explain your figures 3 and 4 in more detail. What are the events in each case? What does the graph show? You should state information like that in the figure legend.

9. In the discussion, you left the text of the MDPI template in there at the beginning (lines 209 to 212). Again, I would like you to refer to my first point made...

10. I do not support your thesis postulated in the section lines 219 to 229. You ultimately reach the conclusion ("Therefore,...") that "patient management and wound treatment are vital for patients". Yet, in the whole section, you write about the chronic and exacerbated pain during wound cleaning and dressing changes, which slowed down patients' life quality. These statement are contradictory in my opinion. Obviously, a chronic wound needs care and the patient needs management, but the premiss does not support the conclusion. I think, this section would be a good place to mention smart and/or advanced wound dressings that try to deal with the drawbacks you mentioned.
Moreover, what wounds are you talking about in this section? The ones of the study patients or chronic wounds in general? Please clarify.

11. All in all, I feel your discussion to be too superficial again. The whole section until line 248 could also be adapted for the introduction. Furthermore, the working mechanisms of PRP falls short again. Important findings have been produced by several studies using PRP recently, e.g. in skin rejuvenation or osteoarthritis - please also refer to these and discuss the working mechanisms. 

12. You merely discuss the results of the studies in ten lines (249 - 259), which I do not feel merits the topic nor your study. Please discuss these eight studies in more detail. What were the main differences? How did they prepare the different dressings? What were the main drawbacks? 

12. Please also discuss an outlook - the potential of PRP? The different available advanced wound dressings? A comparison between them and PRP dressings? 

Thank you.

Author Response

Thank you for your comments concerning our manuscript entitled “The efficiency and safety of platelet-rich plasma dressing in treatment for chronic wounds: a systematic review and meta-analysis of randomized controlled trials” Thank you very much for your comments on our manuscript! The comments are very valuable and helpful for revising and improving our manuscript. We have carefully handled these comments and made some corrections, hoping to get approval. The corrections and the responses to the reviewer’s comments are as follows:

 Comments 1:Your first part of the introduction is just the abstract copied in there (until line 46). I recommend to properly read your manuscript before you submit as as mistake like this rather appears careless showing a certain kind of disrespect towards editorial team and reviewers.

Response and revision 1: Thanks for your kind reviews and comments. We have deleted related content in the manuscript.

Comments 2: Your introduction is rather superficial. You write about growth factors and PRP but on a very superficial basis. Given that the journal is not a specific journal about these two (and even then), I would prefer to read more about current standards, postulated working mechanisms, other indications for the use of PRP and so on.

Response and revision 2: Thanks for your kind reviews and comments. We have added related content to the manuscript. Please see the text highlighted in red in the manuscript (Line 71-84).

The text is in the following:

“As we all know, PRP could release various biological active factors and adhesion proteins to the microenvironment, which contributed to initiating hemostatic cascade, vasculari-zation, and tissue regeneration[18, 19]. Many studies had confirmed that growth factors derived from PRP were able to shorten the wound healing process via the supraphysi-ological releasing of growth factors promoting cell proliferation, migration, and vascu-larization[20]. The neovascularization was the essential process to reconstruct blood supply and support the high metabolic activity of tissue regeneration. In animal ex-periments, it was found that the senescent stem cells could recover proliferation and colony formation ability after PRP treatment, confirming that PRP could resist cell se-nescence during tissue regeneration[21]. Besides, when PRP was exposed to the wounds, the platelets, the main ingredients in PRP, were released as platelets lysate by freeze-thawing disruption[22, 23]. The platelet aggregation was activated, and they would lead to a cascade reaction of cytokines, producing the amount of pain-modulating 5-hydroxytryptamine to release local pain[24].”

Comments 3: Furthermore, you write that traditional treatment included ... the use of "normal saline dressing"... While this might be true to some extent, many new and other wound dressings have emerged that could be used effectively in chronic wounds (some more and some less) in recent years and decades. I recommend to screen the literature again to show some examples and to introduce new insights in wound healing, e.g. about pH and lactate.

Response and revision 3: Thanks for your kind reviews and comments. This is a limitation of this study, so we added related content to the manuscript. Please see the text highlighted in red in the manuscript (Line 53-63).

The text is in the following:

“In order to improve the healing of the chronic wound, several kinds of functional dressing were also used to treat chronic wounds in clinical treatment. Like hydrogel dressing and antimicrobial dressing. The hydrogel dressing could absorb the exudate of the wound and keep it moist, providing an appropriate environment for wound healing, but it was expensive for long-term treatment and lack abundant clinical evidence[11]. The iodine or silver-based dressing had been proven that they had antimicrobial ability in wound treatment, but some studies are concerned that the high concentration of metallic ions would harm the human body, and long-term usage might delay healing[12]. There were also some biomaterial dressings used in chronic wound management, processing bio-compatibility and biodegradability and antimicrobial property, but lacking enough clinical proofs[13].”

Comments 4: In the methods section: I do not understand how you retrieved unpublished studies (line 83) - are you talking about own case series? Or non-reviewed studies published online? Either way, I recommend to NOT include these in your review as this significantly dilutes your results lowering the quality. Same goes for "eligible studies were searched via Google search engine" as Google search (not even google scholar) is not an appropriate means to find sound and peer-reviewed scientific information.

Response and revision 4: Thanks for your kind reviews and comments. We have deleted related content in the manuscript.

Comments 5: Have you registered your review on PROSPERO? If so, please indicate.

Response and revision 5: Thanks for your kind reviews and comments. We have registered this review on INPLASY.COM. The registration number is INPLASY202320053. The DOI number is 10.37766/inplasy2023.2.0053.

Comments 6: You report no heterogeneity between studies reporting a 12-week outcome (line 188), yet the I2 reaches 70% - in my understanding, this heterogeneity is to be considered rather high...? Please comment on that.

Response and revision 6: Thanks for your kind reviews and comments. We have revised the manuscript according to the suggestions. Please see the text highlighted in red in the manuscript (Line 199-200)

The text is in the following:

“There was significant heterogeneity between the two groups (P = 0.03, I2 = 70%).”

Comments 7: Two RCTs reported the healing rate at 8 weeks, and three at 12 weeks... What about the other 3? Did they not report any healing rate at all? If so, please state - I would also like to know the pooled healing rate of all studies. Since this is the main outcome of your study, I think this is of high relevance.

Response and revision 7: Thanks for your kind reviews and comments. We have added relative content to the manuscript according to the suggestions. Please see the text highlighted in red in the manuscript (Line 288-319)

The text is in the following:

“The other included studies also reported consistent outcomes at different endpoints. In Senet et al.[30] study, they found there were no significant differences in the healing rate every day and complete healing rate between the two groups, while PRP dressing group showed a trend of greater healing rate than normal saline dressing group. Elsaid et al.[37] reported the complete healing rate at 20 weeks, the PRP dressing group was significantly higher than the normal saline dressing group ( 3/25 vs 0/25, P=0.03 ), and they showed the time of wound to maximum healing was significantly shorter in PRP dressing group than normal saline dressing group ( 6.3 ± 2.1 vs 10.4 ± 1.7 weeks, P <0.0001 ). Elsaid et al.[37] also found the inefficient rate was significantly lower in PRP group, only 8% of wounds presenting no response to PRP dressing, while more than 66% of wounds having no response to saline dressing. Moneib et al.[35] reported the complete healing rate at end of the study, the PRP group was significantly higher than the normal saline group ( 7/20 vs 0/20, P=0.04 ), and they also recorded the linear healing rate of two groups, PRP group showed a significantly higher healing rate ( P < 0.05 ). Manuel et al.[34] reported the percentage of the healed area after 24 weeks treating, they found the PRP dressing group was significantly quicker than the normal saline dressing group ( 67.7% ± 41.54 vs 11.17% ± 24.4, P = 0.001 ), and the PRP group had significant pain reduction ( P = 0.001 ). Elgarhy et al.[36] evaluated the inflammation and regeneration of chronic wounds via histologic staining. They found local tissues presented less inflammatory cell infiltration and well-formed granulation tissues after six weeks PRP dressing treatment, while moderate vascular proliferation and marked chronic inflammatory cells after six weeks of saline dressing treatment. The results from these RCTs were unanimous, that the PRP dressing could significantly promote chronic wound healing, compared with saline dressing. In addition, some researchers found the chronic wound in the PRP group had bright red granulation after 4 weeks of treatment, which was helpful to reduce inflammation exu-dation[61]. Pires et al.[62] found that there was a similar amount of bacteriological growth in the excretion culturing of PRP and normal saline dressing group. These results con-firmed the mechanisms of PRP, that growth factors from PRP could reduce local in-flammation, and promote angiogenesis and tissue regeneration around the wound.”

Comments 8: Please explain your figures 3 and 4 in more detail. What are the events in each case? What does the graph show? You should state information like that in the figure legend.

Response and revision 8: Thanks for your kind reviews and comments. We have revised the manuscript according to the suggestions. Please see the text highlighted in red in the manuscript (Line 204-206, 222-224).

The text is in the following:

“Forest plot showing the healing rate of PRP dressing versus normal saline dressing. A. The complete healing rate of PRP and control groups at 8 weeks. The events were the number of complete healing wounds at 8 weeks. B. The complete healing rate of PRP and control groups at 12 weeks. The events were the number of complete healing wounds at 12 weeks.”

“Forest plot showing the rate of complications of PRP dressing versus normal saline dressing. A. Infection rate during treatment. The events were the number of infected wounds after treatment. B. Adverse events rate during treatment. The events were the number of adverse events after treatment.”

Comments 9: In the discussion, you left the text of the MDPI template in there at the beginning (lines 209 to 212). Again, I would like you to refer to my first point made...

Response and revision 9: Thanks for your kind reviews and comments. We are very sorry for making this simple mistake. We have delated relative content in the manuscript.

Comments 10: I do not support your thesis postulated in the section lines 219 to 229. You ultimately reach the conclusion ("Therefore,...") that "patient management and wound treatment are vital for patients". Yet, in the whole section, you write about the chronic and exacerbated pain during wound cleaning and dressing changes, which slowed down patients' life quality. These statement are contradictory in my opinion. Obviously, a chronic wound needs care and the patient needs management, but the premiss does not support the conclusion. I think, this section would be a good place to mention smart and/or advanced wound dressings that try to deal with the drawbacks you mentioned.

Moreover, what wounds are you talking about in this section? The ones of the study patients or chronic wounds in general? Please clarify.

Response and revision 10: Thanks for your kind reviews and comments. We have revised the manuscript according to the suggestions. Please see the text highlighted in red in the manuscript (Line 259-261).

The text is in the follows:

“Therefore, finding an available treatment to promote wound healing was necessary for patients suffering from chronic wounds.”

Comments 11: All in all, I feel your discussion to be too superficial again. The whole section until line 248 could also be adapted for the introduction. Furthermore, the working mechanisms of PRP falls short again. Important findings have been produced by several studies using PRP recently, e.g. in skin rejuvenation or osteoarthritis - please also refer to these and discuss the working mechanisms.

Response and revision 11: Thanks for your kind reviews and comments. We have revised the manuscript according to the suggestions. Please see the text highlighted in red in the manuscript (Line 264-283).

The text is in the following:

“Among these treatments, PRP is an advanced method because of the pleasant effect, simple procedure, low cost, and safety[25]. PRP consists of large amounts of growth factors and cytokines, they are released on the wound and can recruit amount of stem cells and various kinds of growth factors via cascade reactions. These cells and factors would con-tribute to regulating epithelial cells proliferation and migration, regulating fibro-blastic[18], promoting angiogenesis and vessel permeability[53], increasing proteins and extracellular matrix synthesis[54]. In vivo, these growth factors could enhance the met-abolic reprogramming of fibroblasts, especially promoting its glycolytic energy metab-olism, to stimulate fibroblasts proliferation and differentiation during tissue repairing[55]. And the concentrated platelets could stimulate the proliferation and pro-angiogenic properties of mesenchymal stem cells even under oxidative stress, to promote angio-genesis and metabolic support around wound[56]. Besides, platelet accumulation would be stimulated by endothelial injury and microbial pathogens in chronic wounds, which could regulate the leukocyte oxidative burst to stimulate the immune response via platelet-neutrophil interactions[57]. The activated immune system could quickly identify and eliminate the viruses and bacteria around chronic wounds, which results in the an-ti-infection effect of PRP dressing[58]. The inflammation would be inhibited after elim-inating infection, and the red granulation tissue formation would be shown at wound[59]. Through these mechanisms, PRP presented promotion in tissue regeneration and chronic wound healing.”

Comments 12: You merely discuss the results of the studies in ten lines (249 - 259), which I do not feel merits the topic nor your study. Please discuss these eight studies in more detail. What were the main differences? How did they prepare the different dressings? What were the main drawbacks?

Response and revision 12: Thanks for your kind reviews and comments. We have revised the manuscript according to the suggestions. Please see the text highlighted in red in the manuscript (Line 290-319).

The text is in the following:

“The other included studies also reported consistent outcomes at different endpoints. In Senet et al.[30] study, they found there were no significant differences in the healing rate every day and complete healing rate between the two groups, while PRP dressing group showed a trend of greater healing rate than normal saline dressing group. Elsaid et al.[37] reported the complete healing rate at 20 weeks, the PRP dressing group was significantly higher than the normal saline dressing group ( 3/25 vs 0/25, P=0.03 ), and they showed the time of wound to maximum healing was significantly shorter in PRP dressing group than normal saline dressing group ( 6.3 ± 2.1 vs 10.4 ± 1.7 weeks, P <0.0001 ). Elsaid et al.[37] also found the inefficient rate was significantly lower in PRP group, only 8% of wounds presenting no response to PRP dressing, while more than 66% of wounds having no response to saline dressing. Moneib et al.[35] reported the complete healing rate at end of the study, the PRP group was significantly higher than the normal saline group ( 7/20 vs 0/20, P=0.04 ), and they also recorded the linear healing rate of two groups, PRP group showed a significantly higher healing rate ( P < 0.05 ). Manuel et al.[34] reported the percentage of the healed area after 24 weeks treating, they found the PRP dressing group was significantly quicker than the normal saline dressing group ( 67.7% ± 41.54 vs 11.17% ± 24.4, P = 0.001 ), and the PRP group had significant pain reduction ( P = 0.001 ). Elgarhy et al.[36] evaluated the inflammation and regeneration of chronic wounds via histologic staining. They found local tissues presented less inflammatory cell infiltration and well-formed granulation tissues after six weeks PRP dressing treatment, while moderate vascular proliferation and marked chronic inflammatory cells after six weeks of saline dressing treatment. The results from these RCTs were unanimous, that the PRP dressing could significantly promote chronic wound healing, compared with saline dressing. In addition, some researchers found the chronic wound in the PRP group had bright red granulation after 4 weeks of treatment, which was helpful to reduce inflammation exu-dation[61]. Pires et al.[62] found that there was a similar amount of bacteriological growth in the excretion culturing of PRP and normal saline dressing group. These results con-firmed the mechanisms of PRP, that growth factors from PRP could reduce local in-flammation, and promote angiogenesis and tissue regeneration around the wound”

Comments 13: Please also discuss an outlook - the potential of PRP? The different available advanced wound dressings? A comparison between them and PRP dressings?

Response and revision 13: Thanks for your kind reviews and comments. We have revised the manuscript according to the suggestions. Please see the text highlighted in red in the manuscript (Line 53-63).

The text is in the follows:

“In order to improve the healing of the chronic wound, several kinds of functional dressing were also used to treat chronic wounds in clinical treatment. Like hydrogel dressing and antimicrobial dressing. The hydrogel dressing could absorb the exudate of the wound and keep it moist, providing an appropriate environment for wound healing, but it was expensive for long-term treatment and lack abundant clinical evidence[11]. The iodine or silver-based dressing had been proven that they had antimicrobial ability in wound treatment, but some studies are concerned that the high concentration of metallic ions would harm the human body, and long-term usage might delay healing[12]. There were also some biomaterial dressings used in chronic wound management, processing bio-compatibility and biodegradability and antimicrobial property, but lacking enough clinical proofs[13].”

Once again, thanks to all reviewers for their valuable reviews and comments!

We really hope that the revisions in the manuscript and our accompanying responses will be sufficient to make our manuscript suitable for publication in Journal of Personalized Medicine. If you have any questions, please do not hesitate to contact me.
 Best wishes,

Fengchao Chen
Medical Cosmetic Center, Beijing Friendship Hospital, Capital Medical University, Beijing 100050, China

Reviewer 2 Report

It is an interesting review study. They aimed to compare with normal saline dressing and the PRP dressing could effectively enhance the prognosis of chronic wounds. The systematic review was conducted according to the Preferred Reporting Items for Systematic reviews and Meta-Analysis (PRISMA) guidelines. Criteria were defined clearly. The number of introduced studies was eight. The only concern about the series included was short follow-up durations. The discussion is short and needs to be improved.

Author Response

Thank you for your comments concerning our manuscript entitled “The efficiency and safety of platelet-rich plasma dressing in treatment for chronic wounds: a systematic review and meta-analysis of randomized controlled trials” Thank you very much for your comments on our manuscript! The comments are very valuable and helpful for revising and improving our manuscript. We have carefully handled these comments and made some corrections, hoping to get approval. The corrections and the responses to the reviewer’s comments are as follows:

Comments 1: The only concern about the series included was short follow-up durations.  

Response and revision 1: Thanks for your kind reviews and comments. We added related content to the manuscript according to the comments. Please see the text highlighted in red in the manuscript (Line 264-283, 347-351).

The text is in the following:

“Among these treatments, PRP is an advanced method because of the pleasant effect, simple procedure, low cost, and safety[25]. PRP consists of large amounts of growth factors and cytokines, they are released on the wound and can recruit amount of stem cells and various kinds of growth factors via cascade reactions. These cells and factors would con-tribute to regulating epithelial cells proliferation and migration, regulating fibro-blastic[18], promoting angiogenesis and vessel permeability[53], increasing proteins and extracellular matrix synthesis[54]. In vivo, these growth factors could enhance the met-abolic reprogramming of fibroblasts, especially promoting its glycolytic energy metab-olism, to stimulate fibroblasts proliferation and differentiation during tissue repairing[55]. And the concentrated platelets could stimulate the proliferation and pro-angiogenic properties of mesenchymal stem cells even under oxidative stress, to promote angio-genesis and metabolic support around wound[56]. Besides, platelet accumulation would be stimulated by endothelial injury and microbial pathogens in chronic wounds, which could regulate the leukocyte oxidative burst to stimulate the immune response via platelet-neutrophil interactions[57]. The activated immune system could quickly identify and eliminate the viruses and bacteria around chronic wounds, which results in the an-ti-infection effect of PRP dressing[58]. The inflammation would be inhibited after elim-inating infection, and the red granulation tissue formation would be shown at wound[59]. Through these mechanisms, PRP presented promotion in tissue regeneration and chronic wound healing.”

 “The outcomes of wounds were irregular, and some data could not be enrolled, so the follow-up durations were short at meta-analysis. We have reported relative outcomes of all enrolled RCTs to increase objectivity. We have tried to include as many patients as possible, but the results may still have some limitations.”

Comments 2: The discussion is short and needs to be improved.

Response and revision 2: Thanks for your kind reviews and comments. We added related content to the manuscript. Please see the text highlighted in red in the manuscript (Line 290-341).

The text is in the following:

“The other included studies also reported consistent outcomes at different endpoints. In Senet et al.[30] study, they found there were no significant differences in the healing rate every day and complete healing rate between the two groups, while PRP dressing group showed a trend of greater healing rate than normal saline dressing group. Elsaid et al.[37] reported the complete healing rate at 20 weeks, the PRP dressing group was significantly higher than the normal saline dressing group ( 3/25 vs 0/25, P=0.03 ), and they showed the time of wound to maximum healing was significantly shorter in PRP dressing group than normal saline dressing group ( 6.3 ± 2.1 vs 10.4 ± 1.7 weeks, P <0.0001 ). Elsaid et al.[37] also found the inefficient rate was significantly lower in PRP group, only 8% of wounds presenting no response to PRP dressing, while more than 66% of wounds having no response to saline dressing. Moneib et al.[35] reported the complete healing rate at end of the study, the PRP group was significantly higher than the normal saline group ( 7/20 vs 0/20, P=0.04 ), and they also recorded the linear healing rate of two groups, PRP group showed a significantly higher healing rate ( P < 0.05 ). Manuel et al.[34] reported the percentage of the healed area after 24 weeks treating, they found the PRP dressing group was significantly quicker than the normal saline dressing group ( 67.7% ± 41.54 vs 11.17% ± 24.4, P = 0.001 ), and the PRP group had significant pain reduction ( P = 0.001 ). Elgarhy et al.[36] evaluated the inflammation and regeneration of chronic wounds via histologic staining. They found local tissues presented less inflammatory cell infiltration and well-formed granulation tissues after six weeks PRP dressing treatment, while moderate vascular proliferation and marked chronic inflammatory cells after six weeks of saline dressing treatment. The results from these RCTs were unanimous, that the PRP dressing could significantly promote chronic wound healing, compared with saline dressing. In addition, some researchers found the chronic wound in the PRP group had bright red granulation after 4 weeks of treatment, which was helpful to reduce inflammation exu-dation[61]. Pires et al.[62] found that there was a similar amount of bacteriological growth in the excretion culturing of PRP and normal saline dressing group. These results con-firmed the mechanisms of PRP, that growth factors from PRP could reduce local in-flammation, and promote angiogenesis and tissue regeneration around the wound.

Besides the healing rate, chronic pain was another main factor decreasing patients’ life quality. During chronic wound management, pain reduction should be a vital clinical goal and managed as early as possible. However, wound pain was repetitive and re-fractory in clinical treatment, and had similar characteristics to chronic pain, like pain centralization and long-term drug use. The mechanisms of chronic pain were complicated, some studies pointed out that the 5-hydroxytryptamine system played vital roles in the modulating magnitude of pain. Many studies tried to find the mechanisms of PRP de-creasing pain, they found PRP could produce the amount of pain-modulating 5-hydroxytryptamine to relieve local pain[24]. In the enrolled RCTs, some studies re-ported that PRP dressing could decrease pain, compared with the normal saline dressing group[34]. Patients’ satisfaction also significantly increased during treatment, following with the pain relieving[34, 36]. While, honestly, PRP was not a strong pain inhibitor, it was hard to achieve the goal of an entirely painless wound after PRP treatment.

In clinical treatment, the infection rate and complications of PRP dressing were se-rious concerns to doctors. According to this study, the infection rate and adverse events were no significant differences between PRP dressing groups and saline dressing groups. Among the included RCTs, there was only one study reporting a case of thrombophlebitis during sampling[30], which did not lead to serious complications. And one study reported two cases of death, one each in PRP and saline dressing group[31], their deaths were unrelated to the chronic wound. Two studies reported that no complications happened during treatment[36, 37]. These results confirmed that PRP dressing would not increase the rate of complications, and it was safe for regularly treating chronic wounds.”

Once again, thanks to all reviewers for their valuable reviews and comments!

We really hope that the revisions in the manuscript and our accompanying responses will be sufficient to make our manuscript suitable for publication in Journal of Personalized Medicine. If you have any questions, please do not hesitate to contact me.
 Best wishes,

Fengchao Chen
Medical Cosmetic Center, Beijing Friendship Hospital, Capital Medical University, Beijing 100050, China
